# Minimum 2-Year Radiographic and Clinical Outcomes of Kinematic Alignment Total Knee Arthroplasty in the Valgus Knee

**DOI:** 10.3390/jpm12071164

**Published:** 2022-07-18

**Authors:** Yaron Bar-Ziv, Eran Beit ner, Konstantin Lamykin, Ahmad Essa, Ron Gilat, Gilad Livshits, Noam Shohat, Yiftah Beer

**Affiliations:** Assaf Harofeh Medical Center, Sackler Medical School, Tel Aviv University, Tel Aviv 6997801, Israel; eranbnster@gmail.com (E.B.n.); klamykin@gmail.com (K.L.); tawohemiz@gmail.com (A.E.); ron.gilat@gmail.com (R.G.); greliv13@gmail.com (G.L.); noam.stam2@gmail.com (N.S.); yiftah.beer@gmail.com (Y.B.)

**Keywords:** kinematic alignment, valgus, patient reported outcomes, clinically significant outcomes, function, arthroplasty

## Abstract

Kinematic alignment (KA) total knee arthroplasty (TKA) has gained popularity in the past decade, but outcomes of KA-TKA in the valgus knee have never been specifically evaluated. In this retrospective single institution study, we analyzed patient reported outcomes and radiographic measurements at minimum 2 years following KA-TKA for valgus knees (*n* = 51) and compared the results to KA-TKA performed for non-valgus knees (*n* = 275). The same approach, technique, and implants were used in both groups without the need to release soft tissues or use constrained implants. Surgery duration was similar between groups (*p* = 0.353). Lateral distal femoral angle was lower in the valgus group postoperatively (*p* = 0.036). In both groups significant improvement was seen in relieving pain and improving function, while average scores were superior in the non-valgus group for visual analog score (*p* = 0.005), oxford knee score (*p* = 0.013), and knee injury and osteoarthritis outcome score (*p* = 0.009). However, these differences did not translate to statistically significant differences in minimal clinical important difference achievement rates. In conclusion, KA-TKA is efficient in relieving pain and improving function, as reported in subjective questionnaires, and holds advantage in patients with valgus alignment by avoiding soft tissue releases and use of constrained implants. Future studies should examine whether bone loss occurs in the lateral distal femur.

## 1. Introduction

Approximately 10% of all primary total knee arthroplasties (TKAs) present with valgus aligned knees [1,2]. The valgus knee creates unique surgical challenges as it often necessitates addressing both bone and soft tissue abnormalities, potentially leading to longer operative time, as well as higher complication and revision rates [1,3,4]. Bone irregularity, contracted lateral capsule, and laxity of medial and posterior soft tissue may create difficulties while balancing a valgus knee during mechanical alignment (MA) TKA [5,6]. Numerous approaches and surgical techniques have been described, but the optimal approach and soft tissue release remain under debate [7,8,9,10,11,12,13].

In recent years, kinematic alignment (KA) for TKA has gained popularity. The main concept of the kinematic approach is replication of the pre-arthritic joint lines and articular surface throughout the knee movement [14,15,16], while avoiding soft tissue release [17,18]. Restoring the pre-arthritic positioning of the bones with minimal soft tissue manipulation results in proper rotation of the femur and tibia, which, in turn, contribute to the integrity of the native soft tissue envelope. Lack of soft tissue manipulation makes KA-TKA a very interesting technique for valgus aligned knees as it simplifies what can become a very complicated operation.

Over the last decade many comparisons between KA and MA for TKA have been conducted, with overall favorable clinical outcomes for KA [17,19]. However, to our knowledge, there has not been a study that specifically reported on the clinical outcomes of caliper-based KA-TKA for the valgus knee. The aim of this study was to evaluate the radiographic and clinical outcomes of KA-TKA for the valgus knee and compare these outcomes to non-valgus aligned knees.

## 2. Materials and Methods

A retrospective, single center study was performed between January 2018 and March 2020 to allow a minimum 2 year follow up. Following institutional review board approval, all consecutive primary TKA cases performed by 3 fellowship-trained surgeons were identified from hospital electronic records. Operative reports were reviewed to verify that KA-TKA was performed. Revision cases were excluded. Patient medical records were extracted and data reporting patient age, body mass index (BMI), comorbidities (using the Charlson Comorbidity Index), type of anesthesia (spinal versus general), operative duration, and length of stay (LOS) were documented. Data on readmissions and reoperations were also extracted and documented.

### 2.1. Technique

Starting January 2018 our institution transitioned from MA-based TKA to calipered KA-TKA using the linked technique. The included surgeons had over 10 years of experience in MA-TKA prior to their transition to KA-TKA in January 2018. The linked technique involves resurfacing the femur using conventional calipered technique which thereafter serves as a guide to cutting the tibia. Shims are used to distract the tibia and achieve soft tissue balance, thus sparing the need for soft tissue release and avoiding cutting into soft bone. After shim placement the knee is examined in extension, flexion, and throughout the range of motion. The goal is to achieve lack of medial opening and to allow slight lateral opening in flexion. Once this goal is achieved, pegs are inserted into the femoral component and the tibial cut is performed parallel to them. Surgeries for valgus and non-valgus knees were performed using the same exact steps with no differences in surgical approach or technique. All surgeries were performed with a medial pivot knee design of the same manufacturer. No stems or constrained implants were used in any of the cases [20].

### 2.2. Radiographic Analysis

The standard protocol at our institution includes EOS imaging at preadmission testing (2–3 weeks prior to surgery), as well as at the first postoperative follow-up visit, two weeks following discharge (Figure 1). Cases in which a preoperative EOS was not available were excluded. Several measurements were performed, including the medial proximal tibia angle (MPTA), lateral distal femoral angle (LDFA), hip knee ankle angle (HKA), and tibial slope. Radiographic analysis was performed by 3 orthopedic residents who were blinded to the clinical outcome assessment. To confirm inter-observer reliability, 20 overlapping cases were examined showing correlation (kappa) of 0.88 (95% confidence interval 0.79 to 0.96).

### 2.3. Follow up Examination

All patients operated on during the above-mentioned period were contacted by phone and mail and were invited to participate in the study. All patients were asked to return for a clinic visit. Those who were not able to attend a clinic visit were phone interviewed by 3 medical students. Waiver from consent was granted for this study as no intervention beyond routine follow-up was performed. Patients were asked to fill the visual analog score (VAS), oxford knee score (OKS), and the Knee Injury and Osteoarthritis Outcome Score (KOOS). Minimal clinical important difference (MCID) for OKS and KOOS were used based on prior literature [21,22]. The MCID for OKS was 5.0. The MCID of the KOOS subscales were 7 (KOOS Symptoms), 18 (KOOS Pain), 16 (KOOS ADL), and 17 (KOOS QOL). Patients were also asked about reoperations associated to the operated joint during their last clinic visit or during phone interview when patients could not attend. Range of motion was also documented at the most recent clinic visit.

### 2.4. Statistical Analysis

The cohort was dichotomized into two categories based on preoperative limb alignment. Patients in whom the center of the knee was lateral to the limb mechanical axis were included in the valgus group. When the mechanical axis went through or medial to the center of the knee, the patients were allocated to the non-valgus group. Comparisons were made between the two groups. Sample size was calculated as 45 patients per group. This was based on a 5-point improvement in the mean OKS from 37 to 42, as previously reported [22,23], with a pooled SD of 8, 80% power, and a two-sided significance level of 5%. Descriptive statistics were calculated for all background characteristics, univariable analysis was conducted using Chi square test for nominal data, interval data were analyzed by T-test for normally distributed data (determined by the Kolmogorov-Smirnoff test) or Mann-Whitney U test (if not normally distributed). The inter-class coefficients (Kappa) were calculated to evaluate the reliability and reproducibility between and within readers. A p-level below 0.05 was considered statistically significant. All analyses were performed using the SPSS packages (version 28.0.1).

## 3. Results

A total of 326 patients were included in the study. Of those, 275 had a neutral or varus alignment preoperatively (non-valgus) and 51 patients had valgus alignment. Time to follow up was 3.34 years (SD 0.79) in the non-valgus group compared to 3.17 years (SD 0.83) in the valgus group (*p* = 0.144). There were no differences between the 2 groups in terms of baseline demographics, comorbidities, or range of motion, except for BMI which was significantly higher in the valgus group (*p* = 0.005). The 2 groups were similar in pain and function prior to surgery (Table 1).

There were statistically significant differences between the two groups in all preoperative radiographic measurements; valgus knees had less varus of the tibia and the femur. Postoperatively, LDFA was decreased (i.e., more valgus) significantly in both groups. In the non-valgus group it changed from 90.10 (SD 3.8) to 84.94 (SD 4.1) and in the valgus group it changed from 86.64 (SD 3.6) to 83.59 (SD 4.40). Consequently, significant differences (*p* = 0.036) remained between the non-valgus (84.94, SD 4.1) and valgus group (83.59, SD 4.40), postoperatively. HKA decreased in both groups postoperatively, and the overall limb alignment remained in varus in the non-valgus group and remained in valgus in the valgus group (Table 2).

While significant improvements in pain and function were seen following surgery (ps < 0.001) in both groups, average VAS, OKS, and KOOS scores were significantly better in the non-valgus group, compared to the valgus group (Figure 2). Average VAS, OKS, and KOOS scores improved by a mean 5.44 (SD 3.33), 25.88 (12.73), and 50.12 (SD 25.50) points in the non-valgus group compared to 4.44 (SD 2.59), 20.72 (SD 10.31), and 46.08 (SD 22.43) in the valgus group (*p* = 0.028, *p* = 0.05 and *p* = 0.275, respectively). KOOS subscales improved for symptoms, pain, function, and quality of life by a mean 40.97 (SD 29.67), 42.90 (SD 31.00), 48.35 (SD 29.06), and 55.39 (SD 28.19) in the non-valgus group compared to 32.56 (SD 25.27), 30.14 (SD 27.16), 39.54 (SD 29.42), and 48.45 (SD 22.59) in the valgus group (*p* = 0.05, *p* = 0.006, *p* = 0.06 and *p* = 0.11, respectively). However, these significant differences did not translate to significant differences in MCID achievement rates between the 2 groups (Table 3). There were also significant differences in postoperative knee extension between the groups which was 1.7 (SD 3.02) in the non-valgus group compared to 5.31 (SD 7.4) in the valgus group. No significant differences were seen in postoperative flexion which was 115.52 (SD 14.48) and 110.31 (SD 14.19), respectively (*p* = 0.179). There were no significant differences in revision rates between the two groups (*p* = 1.00); two cases in the non-valgus group required revision for periprosthetic joint infection and patellar dislocation (0.7%). There were no revision cases in the valgus group (0%).

## 4. Discussion

The main findings of the present study were that KA-TKA was efficient in relieving pain and restoring function (as reported in subjective questionnaires) in patients with valgus alignment, without needing to change approach or technique, or needing to use constrained implants. While patients with a preoperative valgus alignment had statistically significant inferior patient reported outcome scores and inferior improvement in scores compared to patients without valgus alignment, these differences did not reach statistically significant differences in MCID achievement rates. LDFA was significantly lower in the valgus group suggesting inadequate restoration of the femoral component.

Similar to prior reports, valgus knee deformities did not represent the majority of TKA patients undergoing KA-TKA in our series, comprising 15% of our cohort [1,2].

Both soft tissue and bony differences are known factors that may create difficulty in TKA for valgus knees [2,6,24,25]. Using the linked technique, we were able to successfully replace valgus aligned knees without the need to manipulate soft tissue. One major advantage of KA-TKA and, specifically regarding the linking of the tibia to the femur, is that the surgical technique does not differ between varus and valgus aligned knees since the tibial cut is guided by soft tissue tension. The ease of this technique was evident by the similar operative duration seen in the two groups and the lack of use of constrained implants. Furthermore, compared to a recent meta-analysis by Luo et al., the improvement seen in the valgus group in the present cohort was greater than that seen for MA-TKA in previous studies [26]. Young et al. compared 50 MA-TKA and 49 KA-TKA patients and found mean OKS scores of 41 (SD 6) and 42 (SD 6), respectively, which were higher compared to scores reported in the present study. However, it should be noted that preoperative OKS in that study were also higher; 21 (SD 6) and 20.3 (SD 6) in the MA-TKA and KA-TKA groups, respectively [23]. If we compare the delta between our study and the one by Young et al., the improvement in OKS was higher in the present study for both valgus and non-valgus groups. Dossett et al. also reported OKS in MA-TKA and KA-TKA patients in two different studies, mean scores for the MA-TKA group were 33 (SD 11.1) and 33 (SD 8.9) which was lower compared to both groups evaluated in the present study [17,27]. Blakeney et al. compared KOOS scores in MA-TKA and KA-TKA and reported a mean value of 61 (SD 18.1) and 74 (SD 17.1), respectively [28]. The KOOS results reported in our study for both valgus and non-valgus groups were superior to these scores. These findings, together with the lack of complications (i.e., revision surgery), support the use of KA-TKA for valgus knees.

While the surgery was found to benefit all patients, patients with preoperative valgus alignment reported less improvement in terms of self-reported outcome scores following the surgery. While perhaps not clinically meaningful, these differences cannot be overlooked and could have resulted from inability to accurately restore the distal femur [3]. While HKA in non-varus knees was corrected through reducing LDFA and MPTA (i.e., less varus on both the femoral and tibial side), the HKA was corrected in the valgus knees only through the tibia side (i.e., more varus) while LDFA not only did not increase, it was found to have decreased postoperatively (i.e., femoral valgus worsened). Caliper-based KA-TKA relies on restoring the joint line based on cartilage loss assessment. One possible reason for the worsening of LDFA in our series could be a result of undersection of the femoral cartilage in the medial side (when worn-worn pattern is used) or inappropriately assessing cartilage loss. Another possible reason for the inferior scores in the valgus group could be dissatisfaction from overall limb alignment (HKA). Valgus knees started with a significant difference in LDFA compared to non-valgus knees, suggesting possible bone loss on the femoral side. If that is indeed the case, compensation of cartilage alone would have resulted in under correction and increased overall valgus. Of note, while this could not be documented with patient reported outcomes, many patients in the valgus group were unsatisfied with their limb alignment. Furthermore, postoperative extension was worse in the valgus group and could have also affected the results. Future studies should examine whether valgus knee bone loss occurs on the femoral side and what are the best ways to compensate for that loss if, indeed, it exists.

This study had several limitations. First, the follow-up time was relatively short, and might not be sufficient to deduce long term outcomes following KA-TKA for valgus knee osteoarthritis. Second, due to our sample size, the valgus group was not subcategorized to compare different levels of valgus. This could have shed light on reasons for why the valgus group may have had inferior average pain and functional scores. Another limitation is the fact that the study involved three very experienced surgeons at one center. This limits the validity of our results and their applicability to the general orthopedic surgeon population. Furthermore, we included in this study patients who were operated on during the learning curve of three very experienced MA-TKA surgeons and this may have resulted in overall lower reported scores However, this should have affected both valgus and varus knees equally. Finally, all surgeries were based on KA principals and we could not directly compare our outcomes to MA-TKA.

In conclusion, patients presenting with valgus knee alignment create unique surgical challenges that can be avoided using KA-TKA. Based on medium term follow-up, excellent outcomes can be expected without increased complications. Valgus knees with pre-operative extreme LDFA should raise the question as to whether lateral bone loss exists, and, in turn, affects surgical planning. Future studies should evaluate femoral bone loss in this group of patients and compare clinical outcomes between different levels of valgus.

## Figures and Tables

**Figure 1 jpm-12-01164-f001:**
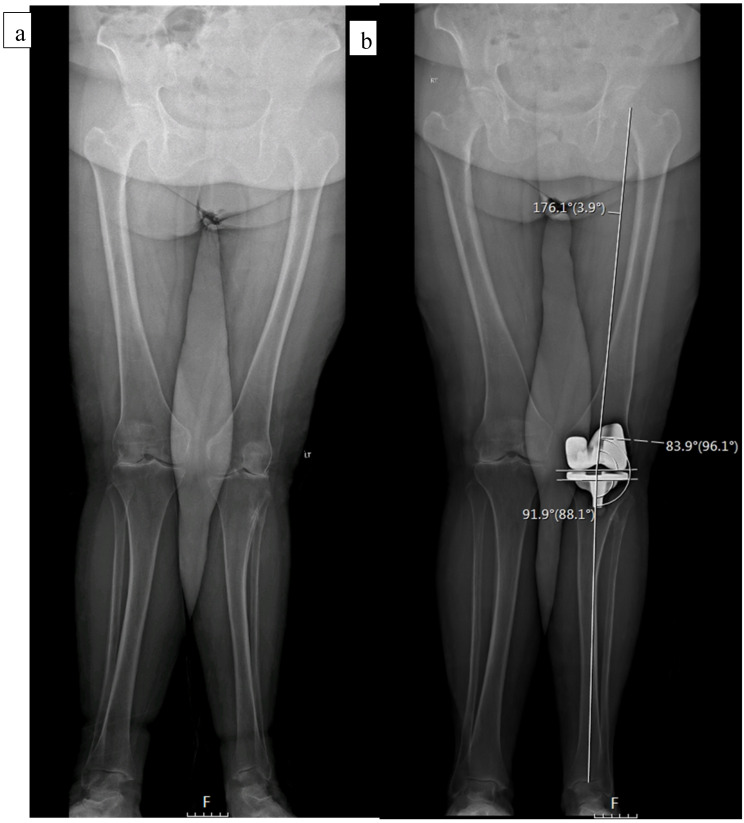
A 76-year-old female with a preoperative valgus alignment (**a**). Postoperative radiographic analysis shows a MPTA of 88.1°, LDFA of 83.9°, and HKA of 3.9° (**b**).

**Figure 2 jpm-12-01164-f002:**
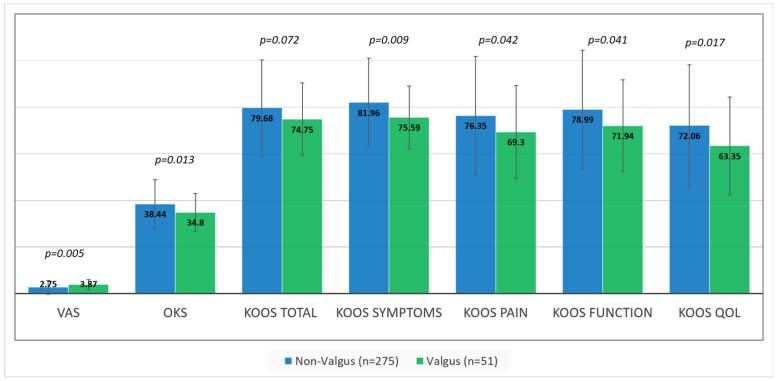
Mean (and SD) patient reported outcome scores in the valgus (green columns) versus non-valgus (blue columns) groups; (KOOS) Knee injury and Osteoarthritis Outcome Score; (OKS) Oxford Knee Score; (VAS) Visual Analog Scale; (QOL) Quality of Life.

**Table 1 jpm-12-01164-t001:** Baseline characteristics, operative factors, and patient reported outcomes in the valgus versus non-valgus groups.

Variable	Non-Valgus (*n* = 275)	Valgus (*n* = 51)	*p*-Value
Age	70.37 (8.08)	70.53 (9.01)	0.905
Gender (female)	172 (62.5%)	39 (76.5%)	0.058
BMI (kg/m)	29.86 (4.95)	32.09 (4.92)	0.005
CCI	0.85 (1.08)	1.15 (1.39)	0.142
Anesthesia (spinal)	204 (74.2%)	32 (62.7%)	0.124
Operative duration	82.63 (19.28)	86.61 (29.02)	0.353
LOS	4.09 (2.24)	4.65 (3.61)	0.292
Extension	4.32 (5.42)	3.53 (4.59)	0.523
Flexion	110.48 (15.38)	106.18 (18.75)	0.378
VAS	8.044 (1.45)	8.314 (1.33)	0.195
OKS	13.40 (7.61)	14.21 (7.84)	0.497
KOOS TOTAL	28.64 (15.10)	28.66 (14.69)	0.992
KOOS SYMPTOMS	40.49 (20.74)	43.02 (22.15)	0.431
KOOS PAIN	31.78 (18.95)	39.16 (17.60)	0.010
KOOS FUNCTION	30.32 (15.28)	32.39 (17.29)	0.385
KOOS QOL	16.32 (8.78)	14.90 (9.10)	0.293
Time to Follow Up (m)	40.18 (9.56)	38.03 (10.00)	0.144

(BMI) Bone Mass Index; (CCI) Charlson Comorbidity Index; (LOS) Length of Stay; (KOOS) Knee injury and Osteoarthritis Outcome Score; (OKS) Oxford Knee Score; (VAS) Visual Analog Scale; (m) months.

**Table 2 jpm-12-01164-t002:** Preoperative and postoperative alignment in the valgus and non-valgus groups.

	Preoperative			Postoperative		
	Non-Valgus (*n* = 275)	Valgus (*n* = 51)	*p*-Value	Non-Valgus (*n* = 275)	Valgus (*n* = 51)	*p*-Value
MPTA	84.33 (3.08)	89.32 (2.74)	<0.001	86.32 (3.11)	87.05 (3.11)	0.127
LDFA	90.10 (3.8)	86.64 (3.6)	<0.001	84.94 (4.1)	83.59 (4.40)	0.036
HKA *	−10.78 (4.79)	5.92 (4.26)	<0.001	−2.83 (2.81)	1.36 (4.18)	<0.001
Slope	11.42 (5.59)	9.03 (4.44)	<0.001	7.01 (4.05)	7.37 (3.15)	0.545

MPTA (medial proximal tibial angle); LDFA (lateral distal femoral angle); HKA (hip knee angle). * Minus represents varus.

**Table 3 jpm-12-01164-t003:** Number and percentage of patients achieving minimal clinical differences (MCID) of the oxford knee score (OKS) and the Knee injury and Osteoarthritis Outcome Score (KOOS) sub-categories in the valgus versus non-valgus groups.

MCID	Non-Valgus (*n* = 275)	Valgus (*n* = 51)	*p*-Value
OKS	262 (95.4%)	48 (93.9%)	0.707
KOOS SYMPTOMS	231 (83.9%)	42 (82.4%)	0.831
KOOS PAIN	215 (78.3%)	35 (68.6%)	0.189
KOOS FUNCTION	236 (85.7%)	42 (82.4%)	0.654
KOOS QOL	243 (88.2%)	46 (90.2%)	0.804

(KOOS) Knee injury and Osteoarthritis Outcome Score; (OKS) Oxford Knee Score; (QOL) Quality of Life.

## Data Availability

The data presented in this study are available on request from the corresponding author. The data are not publicly available due to privacy and ethical reasons.

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
