# Peer review of "Minimum 2-Year Radiographic and Clinical Outcomes of Kinematic Alignment Total Knee Arthroplasty in the Valgus Knee"

_jpm, 2022, doi:10.3390/jpm12071164_

Round 1

Reviewer 1 Report

Thank you for the opportunity to review this manuscript. While the manuscript is well-written and of sound quality, I have a number of questions and suggestions for improvement.

Title

·      I suggest the authors consider amending the title, as the term ‘functional’ suggests the authors conducted quantitative functional assessments on the study participants. Perhaps ‘clinical’ would be more appropriate.

Abstract

·      Please correct ‘visual analyze score’ to ‘visual analogue score’

·      Consider amending the conclusion to clarify that the improvement in function is perceived improvement, as knee function was largely reported in subjective questionnaires.

Materials and Methods

·      What is the level of experience of the 3 surgeons who performed all operations? This is important, because there is likely to have been a learning curve involved in moving from MA-TKA to KA-TKA in 2018. The authors should consider the impact that this learning curve may have had on patient outcomes, and consequently the results presented in this study. This should be presented somewhere in the manuscript.

·      I am not sure what the term ‘queried’ means in the context of the first paragraph. Presumably this does not mean that the authors picked which cases they wanted to include in the study, as this would be extremely biased.  Was there a strict inclusion/exclusion criteria? This should be clarified.

·      Please move the reference to the linked technique to the end of the paragraph. The manuscript should clearly describe the methodology used and not refer readers elsewhere.

·      Please clarify in the follow-up methods whether patients were asked to return to the clinic for a routine follow-up, or whether they were asked to return specifically for the study. This is currently very unclear.

·      Please quote the MCID for the OKS and KOOS in the paper rather than referring the reader elsewhere.

·      It appears that patients were asked by the researchers about their readmission and reoperation rates. When did this happen and how? Similar to my previous comment, this suggests that patients were seen specifically for the study. If this is the case, please include in the methods how patients were approached for the study and how they were consented for the study. Please also explain why these variables were taken directly from the patient and not from their medical records – is collecting this information from patients reliable?

·      How was range of motion measured? The results are presented to the precision of 2 decimal points, which suggests a very sensitive tool was used.

·      Was a sample size calculation performed for this study? How do you know whether your study is powered adequately if not?

·      Please state the level of significance used for the statistical analyses.

Results

·      In what way did the LDFA change in both groups? Please specify.

·      You state that the overall limb alignment remained in varus in the non-valgus group and remained in valgus in the valgus group. What alignments were you aiming for in each group? This should be presented.

·      I could not see Table 2 properly, so could not review the full content of the table. Please reformat so that the table can be adequately reviewed.

·      Only the KOOS Total score is presented in Table 1, but all sub scores are presented in Figure 2 (post-operatively). Please include all scores in Table 1, as it is currently not possible for readers to see how each individual score has changed post-operatively. It may be that the change in score for some scores is consistent in both groups, but that others are not. This could affect the interpretation of the results.

·      As you have not included the MCID values in the methodology, it is currently difficult to know how the scores changed in terms of clinical significance. While the percentages of patients who achieved MCID per score is useful, it would be more valuable to see whether the mean change observed in the scores has achieved clinical significance. Perhaps this could be incorporated into Table 3.

·      Were any other complications reported following TKR? The Results and Discussion only refer to revision rates, but the methods suggests that patients were asked to disclose all complications.

Discussion:

·      Please clarify in the first paragraph that the improvement in function is perceived improvement, as knee function was largely reported in subjective questionnaires.

·      You may want to highlight that while the differences in patient reported outcome measures, the mean scores did improve with statistical and clinical significance.

·      There is no reference to the range of motion results in the discussion.

·      How do the findings of this study relate to the current literature? For example, are the improvements seen in the PROMs typical of what is usually seen? The discussion is currently lacking in this information.

·      The abbreviation PROMs is used for the first time in the discussion without reference. I suggest either using this abbreviation throughout the manuscript and ensuring it is referenced appropriately at first use.

Figures and Tables

·      Is it possible to show the radiographic variables that are presented in the legend on the actual image presented? Please also consider removing the PROMs from the legend, as they are not required here.

·      I suggest highlighting the statistically significant p-values in the tables so that they stand out to readers.

·      Please amend the legend for Figure 2 as it is currently unclear.

Reviewer 2 Report

This study looks at outcomes a KA Valgus knee vs KA non Valgus knees.

The comparison should be KA vs MA Valgus knees (although they have listed it as their limitation and they have stated they do all KA knees these days). What is the logic of comparing Vagus and non Valgus KA knees.

I do not think they can get this data as they no longer do MA TKRs.

Round 2

Reviewer 2 Report

I do not think the authors can change their paper much. I am concerned about comparing KA valgus and KA non valgus knees.

Since the institute does not do any MA knees therefore they will not be able to compare KA and MA. I think this paper is okay and can be published.

Thank you and best wishes.